# Psychometric properties of the Functional Literacy Questionnaire among Portuguese adolescents

**Raquel Martins**[1]*, **Carolina Capitão**[1], **Rodrigo Feteira-Santos**[1], **Ana Virgolino**[1], **Osvaldo Santos**[1,2]

1 Environmental Health Behaviour Lab, Institute of Environmental Health, Lisbon School of Medicine, Lisbon, Portugal, 2 Unbreakable Idea Research, Cadaval, Portugal

* rfmartins@medicina.ulisboa.pt

## Abstract

### Background

Robust tools to assess self-reported adolescent functional health literacy are lacking. In Portugal, the only available tool is the Newest Vital Sign for Portuguese adolescents (NVS-PTeen), though presenting modest validity and reliability properties. A new instrument–the Functional Literacy Questionnaire (FLiQ)–was developed, inspired by the NVS-PTeen, but following the European Regulation for food labeling and targeting a balanced assessment of numeracy and verbal comprehension skills. This study aimed to evaluate several psychometric properties of the FLiQ when administered to Portuguese adolescents.

### Methods

We conducted a longitudinal observational study with three phases: (1) Delphi panel with health literacy experts; (2) self-administration of FLiQ and NVS-PTeen to adolescents in 7th to 9th grades; and (3) re-administration of FLiQ four weeks after baseline, to the same group of participants.

### Results

FLiQ's content validity was excellent, with an Average-Content Validity Index of .95. Overall, 372 adolescents (50.3% girls) aged between 12–17 years (median age: 13) participated in the study. Of these, 150 completed the test-retest assessment. Internal consistency was good (Kuder-Richardson Fornula-20 = .70), as well as test-retest reliability (Intraclass Coefficient Correlation = .82). FLiQ total score was weakly correlated with the school year (rho = .174), and moderately with Portuguese (rho = .348) and Mathematics grades (rho = .333). Factor analysis indicated a two-dimension structure, reflecting numeracy and verbal comprehension skills. Item response theory analysis revealed differences in difficulty and discrimination capacity among items, all with adequate fit values.

**Data Availability Statement:** All relevant files are available on Zenodo (10.5281/zenodo.10696734 for the questionnaire and related information and 10.5281/zenodo.10696710 for the dataset).

**Funding:** This work was supported by funds from Fundação para a Ciência e a Tecnologia (grants UIDB/04295/2020 and UIDP/04295/2020). The funding entity had no role in the research design nor in the writing of this article.

**Competing interests:** The authors have declared that no competing interests exist.

## Conclusion

FLiQ is a valid and reliable tool. It can be used to monitor functional health literacy levels in Portuguese adolescents.

## Introduction

The health literacy concept was introduced in the 1970s [1] and has evolved globally as a main health determinant. Initially, seen as the individual's ability to read and understand health-related information, it has quickly expanded toward a multidimensional construct, which can be described as '*people's knowledge, motivation and competencies to access, understand, appraise and apply information to make judgements and take decisions in everyday life concerning healthcare, disease prevention and health promotion to maintain and improve quality of life during the life course*' [2]. Health literacy encompasses two main sets of skills, namely the ability to perform arithmetic operations and use quantitative information (numeracy) and the ability to read, understand, and locate textual information (verbal comprehension) [2, 3]. According to Nutbeam's model, health literacy is divided into three levels: functional (basic reading and writing skills), interactive (ability to interpret and apply new information), and critical (advanced cognitive and social skills for analyzing and using information required to adequately control life events) [4].

Improving health literacy is a priority in political health agendas worldwide. The European Health Literacy Study from 2019–2021 revealed that 46% of the adult European population had 'inadequate' or 'problematic' general health literacy [5]. In Portugal, the prevalence of limited health literacy decreased from 61% in 2016 [6] to 30% in 2019 [5], which may be a result of the nationwide strategies to promote citizens' health literacy. Nevertheless, these results are based on self-perceptions and may not accurately reflect real health literacy capacities.

Numerous studies have shown that low health literacy is associated with negative health-related outcomes in adults, including non-adherence to preventive behaviors, avoidable hospitalizations and emergency care use [7], decreased quality of life [8], and increased mortality risk [9]. Although fewer studies focus on adolescents, existing evidence suggests that higher parental health literacy is linked to children's healthier nutrition, regular tooth brushing, and more physical activity [10]. Adolescence is a critical period for health literacy promotion due to the significant physical, cognitive, and emotional development that occurs in this life stage [11]. Since health literacy is acquired in a lifelong learning process [12], interventions during childhood and adolescence are pivotal for fostering healthy development and improving long-term health outcomes [13, 14].

Nevertheless, adolescent health literacy remains under-researched, due to the variation of health literacy definitions [2, 15] and the lack of robust assessment instruments [15, 16]. In the last decades, several instruments have been adapted from adult-population questionnaires to assess adolescent health literacy, such as the Test of Functional Health Literacy in adolescent population (TOFHLAd) [17], the Rapid Estimate of Adolescent Literacy in Medicine (REALM-Teen) [18], and the Newest Vital Sign (NVS) [19]. However, these instruments have shown weak to moderate psychometric properties [20], and two of them were not originally designed to comprehensively measure health literacy–TOFHLAd focuses on reading and completing passages [17] and REALM-Teen evaluates the ability to read medical words [18]. In Portugal, the only available tool is the Newest Vital Sign for Portuguese adolescents (NVS-PTeen) [21], which consists of an ice cream nutrition label with six questions measuring

numeracy (four items, all open-ended) and verbal comprehension (two items, one of them close-ended) skills. Santos *et al*. reported an acceptable (not good) internal consistency (Kuder-Richardson Formula-20, KR-20 = .61) and temporal reliability (Intraclass Correlation Coefficient, ICC = .61) properties of this tool [21]. The same authors pointed out that NVS-PTeen is identical to the adult version (only differing from it by adopting the second-person singular) and uses an American nutrition label (not adapted to the Portuguese context). These limitations reinforced the need for additional efforts to develop an adolescent-cultural-adapted scale [21].

To address these issues, we developed the Functional Literacy Questionnaire (FLiQ) to evaluate functional health literacy among Portuguese adolescents. The FLiQ, inspired by the NVS-PTeen, comprises a yogurt nutrition label (information-stimulus) and eight open-ended items assessing numeracy (first four items) and verbal comprehension skills (remaining four items). The main differences between FLiQ and NVS-PTeen are: (a) the information-stimulus regards a food item that is more nutritionally adequate (yogurt, instead of the ice cream presented in the NVS-PTeen); (b) the nutrition label format follows Regulation No. 1169/2011 of the European Union [22] (aiming for greater ecological validity); and (c) the number of items assessing numeracy is the same as the number of items assessing text interpretation skills (four for each dimension, instead of four plus two, respectively for the NVS-PTeen).

Robust health literacy scales can provide valuable information about the relationship between literacy, behaviors, and health outcomes [4]. The aim of this study was to evaluate psychometric properties–validity and reliability–of the FLiQ among Portuguese adolescents enrolled in 7th to 9th grades, using two complementary approaches:

- Classical test theory (CTT): to assess scale's reliability (internal consistency and temporal reliability) and different aspects of validity (content, convergent, concurrent, and construct validity).

- Item response theory (IRT): to examine FLiQ's item performance, focusing on item discrimination, difficulty, and fit.

Combining classical with modern psychometry provides a comprehensive evaluation of the FLiQ's reliability and validity. By employing CTT, this study aims to ascertain the extent to which the FLiQ effectively measures functional health literacy in the target population. As part of CTT, exploratory factor analysis was used to identify the underlying factor structure of the FLiQ. This step is crucial in the development of a new instrument, as it explores the potential dimensions and relationships between items without imposing a predefined structure, helping to ensure that the factors identified are data-driven and representative of the construct being measured [23]. The IRT further refines the evaluation by identifying items with varying discriminative power and difficulty, thereby enhancing the instrument's precision and accuracy.

## Materials and methods

### Study design

The evaluation of the psychometric properties of the FLiQ followed a longitudinal observational study design with three phases: (1) Delphi panel with health literacy experts to characterize FLiQ's content validity; (2) self-administration of the FLiQ and the NVS-PTeen to adolescents enrolled in 7th to 9th grades; and (3) re-administration of the FLiQ to the same group of participants, four weeks after baseline. The NVS-PTeen was chosen for comparison with the FLiQ (concurrent validity) since it was the only available instrument for the Portuguese context that assessed the same construct, i.e., functional health literacy.

## Phase 1

FLiQ's content validity was assessed through a Delphi panel with a group of experts on health literacy, nutrition, adolescents' health, and public health. Experts were chosen based on their professional experience (number of years in function and assigned roles; information available on their curriculum) and scientific productivity in the health literacy field (number of publications in Q1 journals and other relevant works; information available on ORCID or ResearchGate). After identification, experts were invited to participate in the Delphi panel via email.

Data were collected through an online form, built on the LimeSurvey® platform. Each FLiQ item was evaluated for relevance, using a four-point Likert-type response scale (1 = not relevant, 2 = somewhat relevant, 3 = relevant, and 4 = very relevant) and clarity, using a three-point Likert-type response scale (1 = not clear, 2 = item needs revision, and 3 = clear). In both response scales, a neutral rating option of "I don't know/I have no opinion" was included, to ensure that the agreements between experts were not due to chance. The experts could also provide additional comments/suggestions for every FLiQ item.

Regarding relevance, the experts' ratings were used to calculate the Content Validity Index (CVI) for each item (I-CVI) and the overall scale (Ave-CVI). The I-CVI was calculated by the number of experts rating an item as relevant (scores 3 or 4) divided by the total number of experts. This index ranges from 0 to 1, and, according to Polit *et al.*, values ≥.78 indicate that items are relevant, between .70–.77 that the items need revisions, and < .70, that the items should be eliminated [24]. The Ave-CVI was computed by adding the I-CVI of all items, divided by the total number of items; values ≥.90 mean excellent content validity [25].

The FLiQ underwent a meticulous review process (comprising two Delphi panel rounds) until experts attained a consensus regarding the relevance and clarity of all items. The Portuguese version of this instrument, as well as a direct translation to English, is available on Zenodo (10.5281/zenodo.10696734).

## Phases 2 and 3

**Sampling and participants.**   Recruitment focused on adolescents enrolled in 7th to 9th grades from five public schools in two regions of Portugal, selected through a convenience sampling method. We employed a census approach within each school, inviting all eligible adolescents. The recruitment process started with the distribution of informed consent forms to parents/legal guardians, containing detailed information about the study's purpose, procedures, and the voluntary nature of participation. Only students who had the consent form signed by their parents/legal guardians and who agreed to participate in the study (informed assent) filled in the questionnaire. Exclusion criteria were not having Portuguese as the primary language and/or having special education needs (associated with cognitive impairment).

There is no consensus about the adequate number of participants required to evaluate the psychometric properties of an instrument. Considering the mean subject-to-item ratio of 28, as suggested by a systematic review of the literature for determining the sample size for validation processes, 250 participants should be recruited [26]. However, estimating a 40% loss rate of participants in the test-retest assessment (to assess the temporal reliability of the scale), the minimum sample size was settled at 350 adolescents.

**Instruments of data collection and procedures.**   Data were collected through self-administered paper-and-pen questionnaires or, in a reduced number of cases (by requested convenience from some schools), through an online form (LimeSurvey® platform). This step occurred between December 2022 and March 2023. Adolescents completed the FLiQ and the NVS-PTeen and provided basic demographic characteristics (sex and age) and school-related information (school year and self-reported Portuguese and Mathematics grades obtained in

the previous semester or academic year). To uphold the longitudinal component of the study, participants were additionally instructed to complete a pre-assigned random individual code, thereby ensuring the correspondence between the first and second administration of the FLiQ.

At the first administration (phase 2 of the study), the FLiQ and the NVS-PTeen were presented (simultaneously) to all adolescents who agreed to participate and had authorization from their legal guardians. To mitigate any potential order effect, half of the sample responded initially to the FLiQ, followed by the NVS-PTeen, while the other half followed the reverse sequence. For those who completed the online questionnaire, the order was randomly defined by the LimeSurvey® platform.

In the second administration (phase 3), only the FLiQ was re-applied, four weeks later, to the participants who responded to the questionnaire in the previous moment (only the subsample for whom follow-up assessment was possible). The temporal gap between the two administrations was strategically established to mitigate any potential learning bias, a relevant consideration, attending to the reduced number of items of the FLiQ.

**Statistical analysis.** Descriptive statistics including median, interquartile range (IQR; $25^{th}$ percentile, p25 – $75^{th}$ percentile, p75), and frequencies were calculated to describe the variables under study. Data normality was assessed using the Shapiro-Wilk test, complemented by the analysis of kurtosis and skewness of the distributions (data were considered normally distributed if skewness was between -1 to +1 and kurtosis between -2 to +2).

Since participants' ages (continuous variable) were not normally distributed, comparisons between sexes were performed using the Mann–Whitney U test. The comparison of the response time to the FLiQ and the NVS-PTeen (separately) with age group and school year was performed using the Kruskal–Wallis test.

On both functional health literacy scales, participants received one point for each correct answer, with the overall score varying from zero to eight (FLiQ) or from zero to six (NVS-PTeen). The protocol for applying the FLiQ and the correction criteria for each item are also available on Zenodo (10.5281/zenodo.10696734). The NVS-PTeen total score was recoded according to the cutoff points proposed by Weiss *et al.* (the authors of the original American version of the scale) [19]: likelihood of inadequate health literacy (0 to 1 correct answers), limited health literacy (2 to 3 correct answers), and adequate health literacy (4 to 6 correct answers). Regarding the FLiQ, the optimal cutoff points for discriminating individuals with different levels of functional health literacy were determined by the Index of Union Method (based on the value of the area under the receiver operating characteristic (ROC) curve). The cutoff point was defined as the value whose sensitivity and specificity were the closest to the value of the area under the ROC curve, while also ensuring a minimal absolute difference between sensitivity and specificity values [27]. The proportion of adequate/inadequate health literacy, according to FLiQ cutoffs, was compared between sexes, school year, and Portuguese and Mathematics grades using chi-square tests. In case of statistical significance, standardized adjusted residuals for the cell percentage of each subcategory were examined, to determine which cell differences contributed to the chi-squared test results. An adjusted residual score >1.96 (or <−1.96) for a given sub-category percentage indicated that they differed significantly from what would be expected if the variables were independent.

To explore FLiQ's internal validity, KR-20 was computed (due to the dichotomous nature of the variable–correct/incorrect answer) [28], as well as inter-item and item-total correlations. A reliability coefficient of .70 and a corrected item-total subscale correlation of .30 or higher were considered good cutoffs for internal reliability [29]. Coefficient McDonald's omega (ω) was also computed since it is more robust to the assumption of essential tau equivalence (i.e., the same true score for all test items, or equal factor loadings of all items in a factorial model) [30]. Test-retest reliability of the FLiQ was evaluated using the ICC; qualitative interpretations

were as follows: poor (ICC < .40), fair (.40–.59), good (.60–.74) or excellent (.75–1.00) [31]. Temporal reliability was also explored by comparing the proportion of correct/incorrect answers to each FLiQ item at the two administrations. McNemar test was used to determine the differences in the proportion of correct/incorrect answers and Kappa statistics to assess reliability [32, 33]. Strength of agreement was classified as poor (Kappa≤.00), slight (.01–.20), fair (.21–.40), moderate (.41–.60), good (.61–.80) or excellent (.81–1.00) [34].

Concurrent validity was assessed by Spearman's correlation coefficient between FLiQ and NVS-PTeen total scores. Convergent validity was evaluated by Spearman's correlation between the FLiQ total score and four theoretically related variables: age, school year, and Portuguese and Mathematics grades obtained in the previous semester or academic year. Construct validity was analyzed using exploratory factor analysis with direct Oblimin rotation. The Kaiser-Meyer-Olkin (KMO) test and Bartlett's test of sphericity were computed to determine the adequacy of the dataset for this analysis. The correlation matrix of all FLiQ items and the average inter-item correlation were checked to assess the strength of the association between items. Eigenvalue values >1 (scree plot) stated the number of factors and items with factor loading ≥.40 were considered adequate.

Finally, a combined model of a two-three parameter logistic IRT model was used to estimate item difficulty, discrimination, and fit. Item difficulty refers to the literacy level needed for 50% of examinees to get an item correct; discrimination regards the capacity of the scale to differentiate participants with high *versus* low functional health literacy; and fit expresses the degree to which observed responses to an item correspond to expected ones (values of fit higher than .80 indicate an adequate item fit) [35]. The choice for a combined model occurred after verifying that a two-parameter model was not suitable for all FLiQ items (items 1 and 8 did not reveal a good item fit with the two-parameter model and were more adequate when applying the three-parameter model).

Statistical analyses were performed with IBM SPSS® Statistics for Macintosh (version 28.0, 2021, Armonk, NY: IBM Corp) and with jMetrik™ (version 4.0.6, Psychomeasurement Systems, Charlottesville, USA), for IRT analysis. The significance level was set at a two-sided value of $\alpha = .05$.

## Ethical considerations

This study was carried out following the Declaration of Helsinki and was approved by the Ethics Committee of the Centro Académico de Medicina de Lisboa (No. 104/22). Authorizations were also obtained from the direction board of each school where data collection took place. Finally, written informed consents were gathered from all parents, as well as a verbal agreement to participate from adolescents. Before study enrollment, the participants were informed about the study aims, the confidentiality of data collected, that their participation was voluntary, and that filling in the questionnaires would not affect school evaluation (teachers would not have access to the results).

For the subsample participating in the test-retest assessment, anonymity was not possible (only pseudo-anonymity). Nevertheless, it was explained that only members of the research team would have access to the data and that none of the information collected would allow the identification of the adolescent nor would individual data be transmitted to any professional of the involved schools.

## Results

### Phase 1

**Content validity.** The FLiQ underwent two rounds of revisions. The first round included eight experts in health literacy, with the following professional backgrounds: sociology (n = 1),

**Table 1. Characterization of the experts' sample in the two rounds of the Delphi panel.**

|  | Round 1 (N = 8) | Round 2 (N = 7) |
|---|---|---|
| **Sex, n (%)** |  |  |
| Female | 8 (100.0%) | 7 (100.0%) |
| **Age (years), median [IQR]** | 42.50 [36.75–51.75] | 39.00 [36.50–50.00] |
| **Professional experience (years), median [IQR]** |  |  |
| Total | 19.00 [14.25–25.00] | 16.00 [13.50–23.50] |
| Health literacy or adolescent health field | 11.00 [9.50–15.25] | 12.00 [9.00–15.50] |
| **Educational level, n (%)** |  |  |
| Bachelor | 1 (12.5%) | 1 (14.3%) |
| Master | 1 (12.5%) | 1 (14.3%) |
| Doctoral | 6 (75.0%) | 5 (71.4%) |

social policy (n = 1), law (n = 1), nutrition with a focus on adolescence and public health (n = 3), psychology (n = 1), and linguistics (n = 1). Table 1 summarizes the main characteristics of experts in both rounds of the Delphi panel.

Based on the experts' judgment at the first round of the Delphi panel, six (out of eight) FLiQ items were considered relevant for evaluating functional health literacy among adolescents (I-CVI≥.78). Four items obtained an I-CVI = 1.00, meaning that all experts considered these questions relevant to assess the construct under study.

Considering the experts' suggestions collected at the first round, FLiQ's ecological validity was improved by replacing the information-stimulus that was presented to adolescents (for being questioned about). Originally, this stimulus consisted of a food label of a chocolate yogurt and was replaced by a strawberry yogurt (a more common option in Portugal). Also, some items were reformulated to enhance clarity.

The modified version of the FLiQ was presented to the experts in the second round of the Delphi (N = 7 experts) and all items were then considered relevant (I-CVI≥.86) and clear for the target population. The Ave-CVI was .95, indicating that FLiQ has excellent content validity.

## Phase 2

**Sample characterization.** A total of 372 adolescents (50.3% girls) with a median age of 13.00 [IQR 13.00–14.00] years participated in the study (Table 2). Regarding Portuguese and Mathematics grades, 53.2% and 40.6% of the sample reported having 'good' or 'very good', respectively, in the previous semester or academic year. All baseline characteristics were similar between sexes, except for the Portuguese grades, in which the percentage of girls having 'very good' ratings (15.8%) was higher than among boys (9.1%). Also, the percentage of girls having 'sufficient' in this subject was lower compared to boys (19.8% *versus* 33.2%). These results highlight the overall good academic achievement of this sample and a difference between sexes in performance in the Portuguese subject.

**Burden for respondents.** Out of the 372 adolescents who participated in the second phase of the study, 112 (30.1%) responded to both health literacy assessment questionnaires in the online format. This administration mode allowed us to estimate the burden of the FLiQ, for respondents, in terms of response time to the questionnaire. The median time to complete the FLiQ was 09:09 minutes [IQR 08:57–11:34], whereas the median time to complete the NVS-PTeen was 06:25 minutes [IQR 06:33–09:18]. The median response time (in minutes) to both questionnaires by age group and school year are discriminated in Table 3. Overall, although there were no significant differences, the response time to the NVS-PTeen seems to

**Table 2. Total sample characterization: Age, school year, and Portuguese and Mathematics grades from the previous semester or academic year by sex.**

| | Total* (N = 372) | Girls (n = 187) | Boys (n = 177) | p-value** |
|---|---|---|---|---|
| **Age (years), median [IQR]** | 13.00 [13.00–14.00] | 13.00 [13.00–14.00] | 13.00 [13.00–14.00] | .923 |
| **Age group, n (%)** | | | | |
| 12 years | 88 (23.7%) | 40 (22.6%) | 46 (24.6%) | .893 |
| 13 years | 125 (33.6%) | 61 (34.5%) | 61 (32.6%) | |
| 14 years | 124 (33.3%) | 61 (34.5%) | 61 (32.6%) | |
| ≥15 years | 35 (9.4%) | 15 (8.5%) | 19 (10.2%) | |
| **School year, n (%)** | | | | |
| 7th grade | 125 (33.6%) | 59 (33.3%) | 64 (34.2%) | .844 |
| 8th grade | 134 (36.0%) | 62 (35.0%) | 69 (36.9%) | |
| 9th grade | 113 (30.4%) | 56 (31.6%) | 54 (28.9%) | |
| **Final grades: Portuguese, n (%)** | | | | |
| Insufficient | 9 (2.4%) | 1 (0.6%)[b] | 8 (4.3%) | **.002** |
| Sufficient | 98 (26.3%) | 35 (19.8%)[b] | 62 (33.2%) | |
| Good | 151 (40.6%) | 77 (43.5%) | 71 (38.0%) | |
| Very good | 47 (12.6%) | 28 (15.8%)[a] | 17 (9.1%) | |
| No information | 67 (18.0%) | 36 (20.3%) | 29 (15.5%) | |
| **Final grades: Mathematics, n (%)** | | | | |
| Insufficient | 32 (8.6%) | 11 (6.2%) | 21 (11.2%) | .484 |
| Sufficient | 126 (33.9%) | 61 (34.5%) | 63 (33.7%) | |
| Good | 99 (26.6%) | 46 (26.0%) | 50 (26.7%) | |
| Very good | 52 (14.0%) | 22 (12.4%) | 28 (15.0%) | |
| No information | 63 (16.9%) | 37 (20.9%) | 25 (13.4%) | |

Bold indicates statistical significance ($p < .05$).

*N includes "I prefer to describe myself in another way" cases (n = 8).

**Comparison between girls and boys. In the analysis for Portuguese and Mathematics final grades, cases without information were considered as missing values. Chi-Square test for categorical variables and Mann–Whitney U test for continuous variables.

[a]Adjusted standardized residuals >1.96, indicating that the subcategory was observed more frequently than expected if the variables were independent.

[b]Adjusted standardized residuals <-1.96, indicating that the subcategory was observed less frequently than expected.

**Table 3. Median response time (in minutes) to Functional Literacy Questionnaire and Newest Vital Sign for Portuguese adolescents by age group and school year, in minutes (N = 112).**

| | FLiQ | p-value* | NVS-PTeen | p-value* |
|---|---|---|---|---|
| **Age group, median [IQR]** | | | | |
| 12 years | 08:29 [05:01–12:46] | .100 | 06:21 [04:15–09:54] | .454 |
| 13 years | 11:13 [07:43–15:06] | | 06:53 [04:50–09:51] | |
| 14 years | 06:38 [04:41–10:39] | | 05:17 [04:07–08:07] | |
| ≥15 years | 08:56 [05:46–17:06] | | 03:20 [02:47–08:46] | |
| **School year, median [IQR]** | | | | |
| 7th grade | 09:16 [05:08–12:49] | .661 | 07:18 [04:17–10:20] | .167 |
| 8th grade | 09:45 [06:09–15:06] | | 06:12 [04:20–07:02] | |
| 9th grade | 08:41 [05:57–11:10] | | 04:52 [03:30–08:09] | |

FLiQ, Functional Literacy Questionnaire; NVS-PTeen, Newest Vital Sign for the Portuguese adolescents.

*Kruskal-Wallis test.

**Table 4. Spearman's rank correlation coefficients inter-item (correct answers) and item-total score of the Functional Literacy Questionnaire (N = 372).**

|  | Item 2 | Item 3 | Item 4 | Item 5 | Item 6 | Item 7 | Item 8 | Total score |
|---|---|---|---|---|---|---|---|---|
| Item 1 | .346** | .422** | .347** | .197** | .137** | .112* | .194** | .619** |
| Item 2 |  | .500** | .273** | .200** | .202** | .248** | .209** | .640** |
| Item 3 |  |  | .312** | .262** | .188** | .264** | .177** | .695** |
| Item 4 |  |  |  | .083 | .104* | .182** | .260** | .548** |
| Item 5 |  |  |  |  | .246** | .263** | .071 | .497** |
| Item 6 |  |  |  |  |  | .227** | .174** | .495** |
| Item 7 |  |  |  |  |  |  | .217** | .547** |
| Item 8 |  |  |  |  |  |  |  | .487** |

*Significant correlation: $p < .05$

**Significant correlation: $p < .01$.

decrease with age (except for the 13-year-old group) and school year. Regarding the FLiQ, the response time was similar between the school years.

**Internal consistency.** FLiQ's internal consistency was good, with KR-20 = .70 [95%CI .66–.75]. KR-20 coefficient values for each item ranged between .64 and .69, indicating that FLiQ's internal consistency would decrease if any of the eight items were deleted. The inter-item (correct answers) Spearman correlation coefficients are shown in Table 4. All correlations were statistically significant, except for the pairs of items 4–5 and 5–8. The pair of items 2–3 recorded the highest correlation coefficient (rho = .500). Regarding the item-total score correlation (an indicator of item discrimination), Spearman correlation coefficients varied between .487 (item 8) and .695 (item 3). The items with the highest correlations with the FLiQ total score were 1, 2, and 3 (all assessing numeracy skills).

Estimates of McDonald's Omega for the FLiQ were also good, with ω = .70 [95%CI .66–.75]. Regarding NVS-PTeen, the internal consistency in the studied sample was moderate, both by KR-20 = .68 [95%CI .62–.73], as well as by ω = .66 [95%CI .62–.73].

**Convergent and concurrent validity.** FLiQ total score was weakly (though significantly) correlated with the school year (rho = .174 [95%CI .071–.274]), and moderately correlated with Portuguese (rho = .348 [95%CI .242–.446]), and Mathematics grades (rho = 0.333 [95% CI .227–.432]). The correlation with age did not reach a statistically significant result (rho = .083 [95%CI -.022–.186]).

When testing concurrent validity, the FLiQ total score was moderately correlated with NVS-PTeen (rho = .631 [95%CI .563–.690]).

**Construct validity.** After confirming the adequacy of the dataset for factorial analysis (KMO of .78 and Barlett's test of sphericity significant, with $\chi^2$ = 430.2, $p < .001$), two factors emerged with eigenvalues above 1 and factor loading above .4. The eigenvalues for these factors were 2.66 and 1.12, explaining 33.2% and 14.0% (respectively) of the total variance observed (Fig 1). According to the conceptual model of health literacy, factor 1 was associated with numeracy skills, while factor 2 was associated with verbal comprehension skills. Item 8, despite conceptually assessing verbal comprehension skills, was associated with factor 1.

### Phase 3

**Temporal reliability.** Of the total sample, 150 adolescents completed the test-retest assessment. The characteristics of participants are summarized in Table 5.

Comparing the total sample with the subsample (i.e., adolescents who answered the FLiQ twice), a lower proportion of boys and students attending the 9th grade was observed in the second administration of the scale.

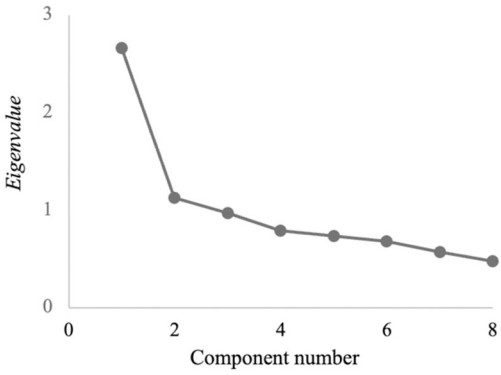

**Fig 1. Exploratory factor analysis.** Results of the exploratory factor analysis after direct oblimin rotation (with the scree plot of eigenvalues) for each item of the Functional Literacy Questionnaire (N = 372).

The test-retest reliability was good, with ICC = .822 [95%CI .755–.871]. Temporal reliability was further assessed by comparing differences in the proportion of correct/incorrect answers between the first and the second administration of the FLiQ. Table 6 shows that from the total of eight questions, only three–items 2, 5, and 8 –presented significant increases in the proportion of correct answers denoting some learning effect. Cohen's Kappa coefficients varied

**Table 5. Subsample characterization: Age, school year, and Portuguese and Mathematics grades from the previous semester or academic year by sex.**

|  | Total (N = 150) | Girls (n = 78) | Boys (n = 72) | p-value* |
|---|---|---|---|---|
| **Age (years), median [IQR]** | 13.00 [12.00–14.00] | 13.00 [12.00–14.00] | 13.00 [12.00–14.00] | .484 |
| **Age group, n (%)** | | | | |
| 12 years | 51 (34.0%) | 27 (34.6%) | 24 (33.3%) | .329 |
| 13 years | 51 (34.0%) | 28 (35.9%) | 23 (31.9%) | |
| 14 years | 39 (26.0%) | 21 (26.9%) | 18 (25.0%) | |
| ≥15 years | 9 (6.0%) | 2 (2.6%) | 7 (9.7%) | |
| **School year, n (%)** | | | | |
| 7th grade | 67 (44.7%) | 35 (44.9%) | 32 (44.4%) | .782 |
| 8th grade | 55 (36.7%) | 30 (38.5%) | 25 (34.7%) | |
| 9th grade | 28 (18.7%) | 13 (16.7%) | 15 (20.8%) | |
| **Final grades: Portuguese, n (%)** | | | | |
| Insufficient | 1 (0.7%) | 0 (0.0%) | 1 (1.4%) | .064 |
| Sufficient | 36 (24.0%) | 14 (17.9%) | 22 (30.6%) | |
| Good | 63 (42.0%) | 31 (39.7%) | 32 (44.4%) | |
| Very good | 27 (18.0%) | 19 (24.4%) | 8 (11.1%) | |
| No information | 23 (15.3%) | 14 (17.9%) | 9 (12.5%) | |
| **Final grades: Mathematics, n (%)** | | | | |
| Insufficient | 5 (3.3%) | 2 (2.6%) | 3 (4.2%) | .646 |
| Sufficient | 49 (32.7%) | 22 (28.2%) | 27 (37.5%) | |
| Good | 45 (30.0%) | 23 (29.5%) | 22 (30.6%) | |
| Very good | 27 (18.0%) | 16 (20.5%) | 11 (15.3%) | |
| No information | 24 (16.0%) | 15 (19.2%) | 9 (12.5%) | |

*Comparison between girls and boys. In the analysis for Portuguese and Mathematics final grades, cases without information were considered as missing values. Chi-Square test for categorical variables and Mann–Whitney U test for continuous variables.

**Table 6. Difference and agreement (correct/incorrect answers) between the two administrations of the Functional Literacy Questionnaire (N = 150).**

| FLiQ 1 | | FLiQ 2 | | | McNemar (*p-value*) | κ [95%CI] |
|---|---|---|---|---|---|---|
| | | Total | Incorrect, n (%) | Correct, n (%) | | |
| Item 1 | Total | 150 (100.0%) | 63 (42.0%) | 87 (58.0%) | 1.000 | .371 [.220–.522] |
| | Incorrect | 63 (42.0%) | 40 (63.5%) | 23 (26.4%) | | |
| | Correct | 87 (58.0%) | 23 (36.5%) | 64 (73.6%) | | |
| Item 2 | Total | 150 (100.0%) | 29 (19.3%) | 121 (80.7%) | **.008** | .274 [.111–.437] |
| | Incorrect | 47 (31.3%) | 17 (58.6%) | 30 (24.8%) | | |
| | Correct | 103 (68.7%) | 12 (41.4%) | 91 (75.2%) | | |
| Item 3 | Total | 150 (100.0%) | 50 (33.3%) | 100 (66.7%) | .268 | .423 [.274–.572] |
| | Incorrect | 58 (38.7%) | 34 (68.0%) | 24 (24.0%) | | |
| | Correct | 92 (61.3%) | 16 (32.0%) | 76 (76.0%) | | |
| Item 4 | Total | 150 (10.0%) | 100 (66.7%) | 50 (33.3%) | .150 | .513 [.366–.660] |
| | Incorrect | 109 (72.7%) | 89 (89.0%) | 20 (40.0%) | | |
| | Correct | 41 (27.3%) | 11 (11.0%) | 30 (60.0%) | | |
| Item 5 | Total | 150 (100.0%) | 40 (26.7%) | 110 (73.3%) | .248 | .564 [.419–.709] |
| | Incorrect | 47 (31.3%) | 30 (75.0%) | 17 (15.5%) | | |
| | Correct | 103 (68.7%) | 10 (25.0%) | 93 (84.5%) | | |
| Item 6 | Total | 150 (100.0%) | 64 (42.7%) | 86 (57.3%) | .371 | .395 [.248–.542] |
| | Incorrect | 71 (47.3%) | 45 (70.3%) | 26 (30.2%) | | |
| | Correct | 79 (52.7%) | 19 (29.7%) | 60 (69.8%) | | |
| Item 7 | Total | 150 (100.0%) | 44 (29.3%) | 106 (70.7%) | **.014** | .448 [.303–.593] |
| | Incorrect | 60 (40.0%) | 33 (75.0%) | 27 (25.5%) | | |
| | Correct | 90 (60.0%) | 11 (25.0%) | 79 (74.5%) | | |
| Item 8 | Total | 150 (100.0%) | 92 (61.3%) | 58 (38.7%) | **< .001** | .583 [.449–.716] |
| | Incorrect | 110 (73.3%) | 87 (94.6%) | 23 (39.7%) | | |
| | Correct | 40 (26.7%) | 5 (5.4%) | 35 (60.3%) | | |

Bold indicates statistical significance (*p* < .05).

FLiQ 1, first administration of the Functional Literacy Questionnaire; FLiQ 2, second administration of the Functional Literacy Questionnaire; κ, Cohen's Kappa coefficient.

between .274 and .583, with items 3, 4, 5, 7, and 8 showing moderate agreement. The remaining items (1, 2, and 6) revealed regular agreement (κ = .371, κ = .274, and κ = .395, respectively).

### Item response theory parameters

Item characteristic curves revealed that items 1, 2, and 3 (a = 2.01, a = 2.09, and a = 2.29, respectively) better discriminated between individuals with adequate *versus* inadequate functional health literacy, while items 5, 6, and 7 had the least discriminating capacity (a = .82, a = .74, and a = .95, respectively; Fig 2). Regarding difficulty, items 2, 5, and 7 were the easiest (b = -0.59, b = -0.71, and b = -0.26, respectively), whereas items 4 and 8 were the most difficult ones (b = 1.09, and b = 1.33, respectively). Values of unweighted mean squares and weighted mean squares showed an adequate fit for all items.

### Functional health literacy levels

Fig 3 shows the ROC curve for the FLiQ and the sensitivity and specificity values for each cut-off point.

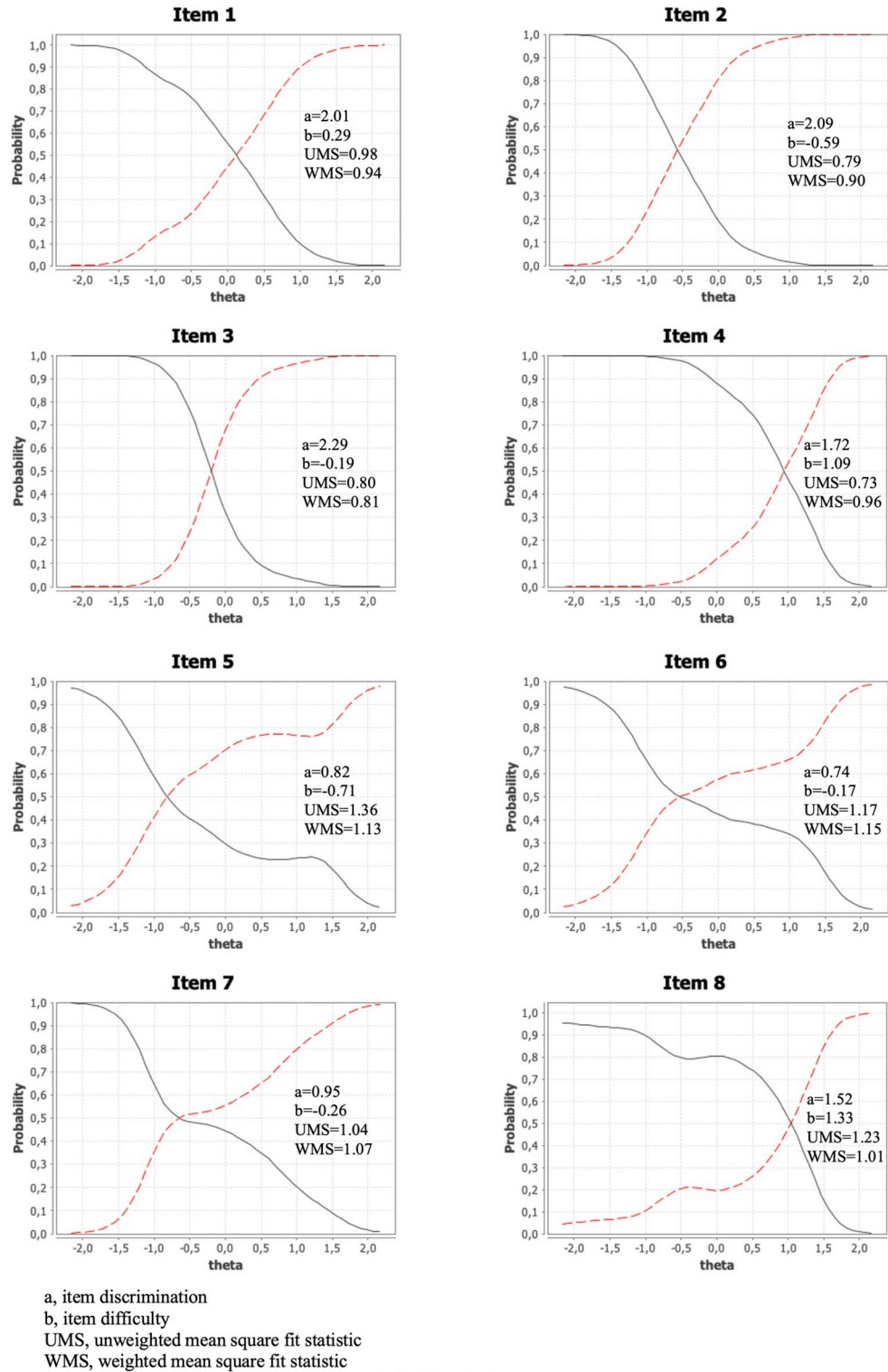

a, item discrimination
b, item difficulty
UMS, unweighted mean square fit statistic
WMS, weighted mean square fit statistic
Continuous line (black) – incorrect answer; dashed line (red) – correct answer

**Fig 2. Item characteristic and information curves (N = 372).** Discrimination and difficulty values for each item of the Functional Literacy Questionnaire.

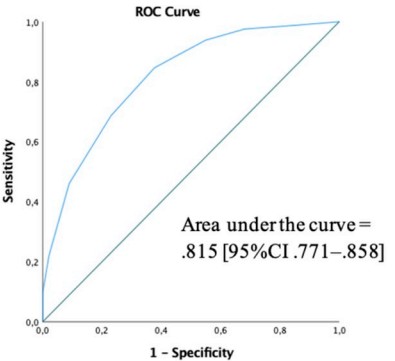

| Cutoff values | Sensitivity | Specificity* |
|---|---|---|
| 0.5 | .988 | .152 |
| 1.5 | .975 | .319 |
| 2.5 | .939 | .450 |
| 3.5 | .847 | .623 |
| 4.5 | .687 | .770 |
| 5.5 | .460 | .911 |
| 6.5 | .221 | .979 |
| 7.5 | .098 | 1.000 |

*Specificity calculated as 1-sensitivity.

**Fig 3. ROC curve.** Sensitivity and specificity values for predicting different cutoff points of the Functional Literacy Questionnaire.

According to the Index of Union method, the value of 4.5 was the optimal cutoff point, given that the respective sensitivity (.687) and specificity (.770) values were the closest to the area under the curve (.815) and, simultaneously, the difference between them was minimal. Therefore, two health literacy levels were considered: 'limited' when the FLiQ total score was <5 points and 'adequate' when the score was ≥5 (out of 8). Since FLiQ aims to evaluate both numeracy and verbal comprehension skills, in addition to a total score ≥5 points, it was defined as an adequate functional health literacy criterion when answering correctly in at least two items of each dimension.

Considering these criteria of the FLiQ (and the cutoffs proposed for the NVS described in the methods section), Table 7 presents the functional health literacy levels of the total sample and stratified by sex, school year, and Portuguese and Mathematics final grades. Overall, 42.5% of the adolescents had adequate functional health literacy according to the FLiQ and 47.3% by the NVS-PTeen, suggesting that FLiQ is a more conservative tool than NVS-PTeen. The proportion of adequate health literacy level, in both scales, was higher in adolescents enrolled in 8th grade, compared to those in the 7th grade. Adequate health literacy measured by the FLiQ was also more frequent among participants who reported having the highest grades in Portuguese ('good' or 'very good') and Mathematics ('very good'), in comparison to those who reported having the lowest grades ('insufficient' or 'sufficient').

## Discussion

This study aimed to characterize psychometric properties of the FLiQ (self-administered) among Portuguese adolescents in 7th to 9th grades. The combination of CTT with IRT allowed a comprehensive understanding of the overall scale and individual item performance in measuring functional health literacy.

Content validity was evaluated using the Delphi method, whose main virtue lies in its ability to reach consensus among experts [36, 37]. Although there are no clear guidelines on the adequate number of experts, Almanasreh *et al.* suggest that a Delphi panel should be composed of five to 10 [38]. Considering this recommendation, the sample size included in both rounds of the Delphi panel was adequate. As for the quantitative analysis, all FLiQ items achieved adequate values of I-CVI (≥.86) and the Ave-CVI was excellent (of .95), indicating that all questions are relevant and clear to evaluate functional health literacy in adolescents.

In terms of time required to completion (a very relevant burden issue), the NVS-PTeen took a median of six minutes, which is consistent with the adult version of the NVS reported in the literature [19, 39]. In our study, the NVS-PTeen took six minutes to complete (median

**Table 7. Comparison of functional health literacy levels assessed by the Functional Literacy Questionnaire and by the Newest Vital Sign for the Portuguese adolescents by sex, school year, and Portuguese and Mathematics grades from the previous semester or academic year.**

| | Functional health literacy | | | | |
| --- | --- | --- | --- | --- | --- |
| | FLiQ | | NVS-PTeen | | |
| | Limited (score <5) | Adequate (score ≥5) | Inadequate (score 0–1) | Limited (score 2–3) | Adequate (score 4–6) |
| **Total\* (N = 372), n (%)** | 214 (57.5%) | 158 (42.5%) | 73 (19.6%) | 123 (33.1%) | 176 (47.3%) |
| **Sex, n (%)** | | | | | |
| Female (n = 187) | 99 (55.9%) | 78 (44.1%) | 34 (19.2%) | 63 (35.6%) | 80 (45.2%) |
| Male (n = 177) | 108 (57.8%) | 79 (42.2%) | 36 (19.3%) | 58 (31.0%) | 93 (49.7%) |
| *p-value*\*\* | .807 | | .617 | | |
| **School year, n (%)** | | | | | |
| 7th grade (n = 125) | 89 (71.2%) | 36 (28.8%)[b] | 42 (33.6%)[a] | 49 (39.2%) | 34 (27.2%)[b] |
| 8th grade (n = 134) | 59 (44.0%) | 75 (56.0%)[a] | 17 (12.7%)[b] | 34 (25.4%) | 83 (61.9%)[a] |
| 9th grade (n = 113) | 66 (58.4%) | 47 (41.6%) | 14 (12.4%)[b] | 40 (35.4%) | 59 (52.2%) |
| *p-value*\*\*\* | < .001 | | < .001 | | |
| **Final grades: Portuguese, n (%)** | | | | | |
| Insufficient (n = 9) | 8 (88.9%) | 1 (11.1%)[b] | 6 (66.7%)[a] | 1 (11.1%) | 2 (22.2%) |
| Sufficient (n = 98) | 70 (71.4%) | 28 (28.6%)[b] | 25 (25.5%)[a] | 40 (40.8%) | 33 (33.7%)[b] |
| Good (n = 151) | 72 (47.7%) | 79 (52.3%)[a] | 18 (11.9%)[b] | 50 (33.1%) | 83 (55.0%) |
| Very good (n = 47) | 19 (40.4%) | 28 (59.6%)[a] | 3 (6.4%)[b] | 8 (17.0%) | 36 (76.6%)[a] |
| *p-value*\*\*\* | < .001 | | < .001 | | |
| **Final grades: Mathematics, n (%)** | | | | | |
| Insufficient (n = 32) | 26 (81.3%) | 6 (18.8%)[b] | 12 (37.5%)[a] | 10 (31.3%) | 10 (31.3%)[b] |
| Sufficient (n = 126) | 82 (65.1%) | 44 (34.9%)[b] | 27 (21.4%) | 55 (43.7%) | 44 (34.9%)[b] |
| Good (n = 99) | 49 (49.5%) | 50 (50.5%) | 12 (12.1%) | 27 (27.3%) | 60 (60.6%)[a] |
| Very good (n = 52) | 18 (34.6%) | 34 (65.4%)[a] | 2 (3.8%)[b] | 8 (15.4%) | 42 (80.8%)[a] |
| *p-value*\*\*\* | < .001 | | < .001 | | |

Bold indicates statistical significance (*p* < .05).

FLiQ, Functional Literacy Questionnaire; NVS-PTeen, Newest Vital Sign for the Portuguese adolescents.

\*N includes the cases "I prefer to describe myself in another way" (n = 8).

\*\*Comparison between girls and boys. Yates's correction for continuity (tables 2x2).

\*\*\* Chi-Square test. In the analysis of Portuguese and Mathematics final grades, cases without information were considered as missing values.

[a]Adjusted standardized residuals >1.96, indicating that the subcategory was observed more frequently than expected if the variables were independent.

[b]Adjusted standardized residuals <-1.96, indicating that the subcategory was observed less frequently than expected.

Notes: Differences between groups regarding total FLiQ score for: school year (7th grade differs from 8th grade), final Portuguese grades ('sufficient' differs from 'good' or 'very good'), and final Mathematics grades ('insufficient' differs from 'good' or 'very good'; 'sufficient' differs from 'very good'). Differences between groups regarding total NVS-PTeen score for: school year (7th grade differs from 8th or 9th grade), final Portuguese grades ('sufficient' differs from 'good' or 'very good'), final Mathematics grades ('insufficient' differs from 'very good'; 'sufficient' differs from 'good' or 'very good'; 'good' differs from 'very good').

response time). The FLiQ took nine minutes; this somewhat longer response time (compared to the NVS-PTeen) was expected due to the additional items and the open-ended format, which require more cognitive processing [40].

FLiQ's internal consistency was good, indicating that all items effectively contribute to measuring different aspects of functional health literacy. Importantly, removing any of the items would decrease the KR-20 coefficient, emphasizing the importance of each individual item to evaluate the construct under study. The study by Santos *et al.* on the psychometric properties of the NVS-PTeen among Portuguese adolescents showed an acceptable, though not so good, internal consistency [21]. Our findings align with the good internal consistency found for the

American [19] and Portuguese [41] adult versions of the NVS. It is important to note that KR-20 and Cronbach's α are sensitive to the number of scale items [42], which is why internal consistency should be complemented with inter-item correlation analysis. In our study, the correlation coefficients of the FLiQ were satisfactory (mostly varying between .20 and .40), mirroring the results of previous studies on adolescents [21] and adults [39].

FLiQ total score was moderately correlated with NVS-PTeen, which supports the presence of a conceptual relationship between the two scales in assessing functional health literacy in adolescents.

About convergent validity, FLiQ total score was weakly associated with the school year and moderately with Portuguese and Mathematics final grades. This finding is particularly relevant since FLiQ assesses functional health literacy based on numeracy and verbal comprehension skills. The absence of a significant association with age suggests that the scale is not linearly affected by age-related developmental biases. Including adolescents of a wider span of age could likely lead to increased health literacy differences through age. Although we expected a higher correlation with Portuguese and Mathematics grades, it is important to note that academic performance is influenced by several factors beyond the scope of our study, such as teaching methods, mental health, time spent on gadgets, family socioeconomic status, and parenteral support [43, 44]. This suggests that improving academic skills can enhance health literacy, highlighting the importance of a multidisciplinary approach in education that integrates health literacy into the curriculum.

Exploratory factor analysis revealed a two-dimension structure: numeracy (items 1, 2, 3, 4, and 8) and verbal comprehension skills (items 5, 6, and 7). This is consistent with the dimensionality reported in other psychometric assessment studies of the NVS for Portuguese populations–both the adolescent [21] and adult versions [39]. Items 5, 6, 7, and 8 of the FLiQ were originally designed to assess verbal comprehension skills; therefore, the results did not entirely align with the predicted theoretical framework. Although item 8 was associated with factor 1 (numeracy skills), it involves semantic equivalence between terms (which, according to the conceptual model of health literacy, is an exercise that assesses verbal comprehension skills); as so, we consider that item 8 should be included in factor 2. Educators and healthcare providers can use FLiQ to screen adolescents' health literacy and identify educational needs. The two-dimensional structure, reflecting numeracy and verbal comprehension skills, provides detailed insights into specific areas where students may struggle, enabling tailored strategies to address these gaps.

Concerning reproducibility, we observed a learning effect between the two administrations of the FLiQ, with deviations toward improved health literacy levels. This phenomenon was also noted by Santos *et al.* with the NVS-PTeen [21]. Despite this, the temporal reliability of the FLiQ was higher than that reported for the NVS-PTeen, suggesting that FLiQ is a more robust tool in reproducing consistent results over time in the same group of participants. Additionally, this result indicates that FLiQ is suitable for evaluating the impact of educational programs or policy changes on health literacy levels over time.

The IRT analysis showed that all items had acceptable levels of discrimination, effectively differentiating between participants with varying trait levels (i.e., functional health literacy). Items 1, 2, and 3 (all assessing numeracy skills) were identified as the most discriminative.

Health literacy is an asset that empowers individuals to exert greater control over their health [13]. Promoting health literacy during adolescence is pivotal, as health-related behaviors established in this life stage are closely linked to health outcomes in adulthood [13]. Despite the efforts in the last decades to develop robust instruments to assess functional literacy among adolescents, this remains a knowledge gap.

To the best of our knowledge, only the study conducted by Santos *et al.* has characterized functional literacy among Portuguese adolescents aged 12 to 17 years [21]. By applying the NVS-PTeen, the authors found that 83.4% of the sample had adequate health literacy levels. This contrasts with the NVS-PTeen data collected in our study, in which less than half of the sample revealed adequate health literacy. This may be due to differences in participant characteristics–our investigation focused only students attending 7th to 9th grades, whereas Santos *et al.* study included about 40% of participants from 10th grade and above [21]. Furthermore, our study revealed higher levels of limited functional health literacy when using the FLiQ compared to the NVS-PTeen. This discrepancy may be attributed to FLiQ's classification system, which requires good performance in both numeracy and verbal comprehension items.

## Strengths and limitations

FLiQ is the first scale to evaluate functional health literacy in adolescents using a food label with greater ecological validity, compliant with Regulation No. 1169/2011 of the European Union. Additionally, the literacy level classification system of FLiQ integrates numeracy and verbal comprehension skills, combined with a cutoff value defined by the sensitivity and specificity properties of the scale. This makes FLiQ a more balanced instrument for assessing health literacy, compared to the NVS-PTeen.

The sample size for the cross-sectional approach was highly satisfactory (N = 372), with a balanced distribution by gender (50.3% girls). However, it is important to note that the loss rate of participants in the test-retest assessment was higher than expected, which could have reduced the statistical power of the longitudinal phase of the study.

Data were collected through two strategies–online and paper-and-pen questionnaires, which could introduce some variety in the responses. Evidence shows that the answers to a questionnaire tend not to differ, regardless of the modality in which they were administered, if maintaining the layout [45]. In our study, it was not possible to keep the same layout between the two administration modes due to the technical specifications of the platform used. However, when performing a sensitivity analysis (removing cases that responded to the questionnaire in the online format), results continued to point out the robust validity and reliability of the FLiQ.

Data on functional health literacy levels should not be generalized to the overall adolescent population, since most participants were recruited from one school. On the other hand, given the strong psychometric properties of the FLiQ, this instrument can be applied to larger and more heterogeneous samples of adolescents. Additionally, policymakers could consider incorporating this new tool into national health education programs, for monitoring the effectiveness of health literacy interventions and guide resource allocation to areas with the greatest need.

## Conclusions

In the last decades, health literacy has gained attention, both in research and practice, due to its association with health behavior and related outcomes. Consequently, there has been a demand for robust tools to assess it, especially among adolescents. This study shows that FLiQ has good psychometric properties, supporting the utility of this instrument to effectively assess health literacy and identify vulnerable groups (to harmful behaviors or less favorable health conditions).

Given the inexistence of a nationwide dataset of functional health literacy levels of adolescents in Portugal, FLiQ could be used as a monitoring tool applied in schools. This nationwide monitoring initiative would allow the identification of knowledge gaps and guide the

development of effective health literacy policies and interventions. FLiQ is also adequate to be used as an outcome indicator of health literacy interventions.

## Acknowledgments

The authors want to express their gratitude to all health literacy experts who participated in the Delphi panel, for their valuable contributions that led to the final version of the FLiQ. A special thanks to the school boards and teachers at all schools where data collection took place. The authors would also like to acknowledge all students and their parents for participating in the study.

## Author Contributions

**Conceptualization:** Raquel Martins, Carolina Capitão, Rodrigo Feteira-Santos, Ana Virgolino, Osvaldo Santos.

**Formal analysis:** Raquel Martins, Osvaldo Santos.

**Methodology:** Raquel Martins, Carolina Capitão, Rodrigo Feteira-Santos, Ana Virgolino, Osvaldo Santos.

**Supervision:** Osvaldo Santos.

**Writing – original draft:** Raquel Martins.

**Writing – review & editing:** Raquel Martins, Carolina Capitão, Rodrigo Feteira-Santos, Ana Virgolino, Osvaldo Santos.

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
