## [Decision Letter · Decision Letter 0]

7 May 2024

PONE-D-24-07888Psychometric properties of the Functional Literacy Questionnaire among Portuguese adolescentsPLOS ONE

Dear Dr. Martins,

Thank you for submitting your manuscript to PLOS ONE. After careful consideration, we feel that it has merit but does not fully meet PLOS ONE’s publication criteria as it currently stands. Therefore, we invite you to submit a revised version of the manuscript that addresses the points raised during the review process.

We look forward to receiving your revised manuscript.

Kind regards,

Maria José Nogueira, Ph.D.

Academic Editor

PLOS ONE

Journal Requirements:

3. Please upload a copy of Supporting Information Figure/Table/etc. "Supporting information" which you refer to in your text on page 35.

**Additional Editor Comments:**

Ethical considerations - Authors must mention the authorization number of the Ethics Committee that approved the study.

It is a relevant study with a robust methodology and statistical treatment, which adds pertinent knowledge in this area.

The references need to be revised in an effort to use more recent authors, as they have a large percentage of articles that are more than 10 years old.

Reviewers' comments:

Reviewer's Responses to Questions

**Comments to the Author**

1. Is the manuscript technically sound, and do the data support the conclusions?

Reviewer #1: Yes

2. Has the statistical analysis been performed appropriately and rigorously? 

Reviewer #1: Yes

3. Have the authors made all data underlying the findings in their manuscript fully available?

Reviewer #1: Yes

4. Is the manuscript presented in an intelligible fashion and written in standard English?

Reviewer #1: Yes

5. Review Comments to the Author

Reviewer #1: GENERAL COMMENTS:

The study aimed to assess the validity and reliability of the Functional Literacy Questionnaire for Portuguese adolescents using CFA. While the topic is intriguing, several issues need addressing before publication.

SPECIFIC COMMENTS:

The review appears limited with only two identified manuscripts. Expanding the search period beyond 2020-2023 and adding keywords could potentially yield more relevant literature and broaden the review's scope.

INTRODUCTION:

Minor revisions are required for clarification in the introduction. Specify the study's aim and establish a clear research question. Restructure the introduction to emphasize the study's significance and rationale behind the CFA investigation. Streamline the introduction for conciseness.

METHODS:

Provide more information on participant recruitment methods for clarity.

Include a psychologist in the panels for Phase 1.

RESULTS:

In Table 2, emphasize the significance of the results by explaining the findings more explicitly.

DISCUSSION:

Expand on the implications of the study findings in the discussion, focusing on result-related discussions. Avoid delving into irrelevant topics and trim unnecessary details of previous studies. Enhance the discussion by considering limitations more comprehensively. Update the reference list with newer sources.

Overall, well done on the study.

Thank you.

6. PLOS authors have the option to publish the peer review history of their article (what does this mean?). If published, this will include your full peer review and any attached files.

Reviewer #1: No

---

## [Author Response · Author response to Decision Letter 0]

21 Jun 2024

Manuscript number: PONE-D-24-07888

Lisbon, 21 June 2024

Dear Editor-in-Chief Dr. Emily Chenette,

Thank you for the opportunity to submit a revised version of our manuscript, “Psychometric properties of the Functional Literacy Questionnaire among Portuguese adolescents”, for publication in PLOS ONE. 

We appreciate the interest shown in our study and the insightful comments that were made. We have carefully revised the manuscript according to the suggestions of the academic editor and the reviewer, which greatly improved the quality of the manuscript. 

Please consider the following answers to the comments/suggestions, one by one. We have also submitted a revised manuscript with changes highlighted, as required.

We look forward to hearing from you soon and we are available to respond to any further questions or comments you may have.

Yours, sincerely,

Raquel Martins

Environmental Health Behavior Lab, Institute of Environmental Health, Lisbon School of Medicine

Avenida Professor Egas Moniz 1649-028 Lisboa, PORTUGAL

Phone: +351 21 799 94 89

E-mail: rfmartins@medicina.ulisboa.pt

Note: Lines and page numbers are referred to the version of the manuscript without track changes.

Reviewers' Comments to the Authors: 

Academic Editor:

Comment: Ethical considerations - Authors must mention the authorization number of the Ethics Committee that approved the study. It is a relevant study with a robust methodology and statistical treatment, which adds pertinent knowledge in this area.

Reply: We appreciate the editor’s acknowledgment of the relevance and robustness of our study. As suggested, we have included the authorization number of the Ethics Committee in the ethical considerations section (lines 273-4, page 12).

Comment: The references need to be revised in an effort to use more recent authors, as they have a large percentage of articles that are more than 10 years old.

Reply: Thank you for your valuable feedback. We have thoroughly revised the reference list to include more recent sources whenever possible to ensure that our manuscript reflects the most current research. We have retained some references that are over 10 years old due to their importance in this field. Those references concern statistical methods, cutoff points, and conceptual definitions that remain widely accepted and have not been superseded by more recent research.

Reviewer #1:

Comment: The review appears limited with only two identified manuscripts. Expanding the search period beyond 2020-2023 and adding keywords could potentially yield more relevant literature and broaden the review's scope.

Reply: We have now conducted a more comprehensive search focusing on the health literacy concept, dimensions and components, its relationship with health outcomes, and the prior application of tools that assess functional health literacy levels. We explored the literature again without time restrictions, including keywords that returned a wide range of results. We have reformulated the introduction to make it clearer and more concise.

Comment: Minor revisions are required for clarification in the introduction. Specify the study's aim and establish a clear research question. Restructure the introduction to emphasize the study's significance and rationale behind the CFA investigation. Streamline the introduction for conciseness.

Reply: Thank you for your comment. We have revised the introduction and streamlined the text for conciseness, as suggested. We also reformulated the aim of the study and explored the rationale for the use of classical test theory (including exploratory factor analysis) and item response theory in this investigation (lines 104-21, pages 5-6). Changes have been made throughout the entire Introduction section (pages 3-6).

Comment: Provide more information on participant recruitment methods for clarity. Include a psychologist in the panels for Phase 1.

Reply: We reviewed the methods section and detailed the recruitment process for experts (lines 135-42; pages 6-7) and adolescents (lines 164-70; page 8). Regarding the composition of the Delphi Panel, we had included a psychologist in the group, however, this was not clearly stated in the manuscript. To address your comment, which we greatly appreciate, we have now specified the professional backgrounds of all participating experts in the results section (lines 289-92; page 13).

Comment: In Table 2, emphasize the significance of the results by explaining the findings more explicitly.

Reply: We have added a sentence highlighting the overall academic achievement of the studied sample and the difference between sexes regarding Portuguese classifications (lines 320-1; page 14). 

Comment: Expand on the implications of the study findings in the discussion, focusing on result-related discussions. Avoid delving into irrelevant topics and trim unnecessary details of previous studies. Enhance the discussion by considering limitations more comprehensively. Update the reference list with newer sources.

Reply: Thank you for your insightful comment. We explored more comprehensively the implications of our findings and the limitations of our study (pages 24-30). We have revised the discussion and trimmed unnecessary details of previous studies. We also have updated the reference list, as indicated previously. We believe these revisions enhance the clarity and relevance of the discussion section.

Journal Requirements:

Reply: We have reviewed PLOS ONE's style requirements and made changes accordingly, namely on the headings (capitalizing only the first word), and the format of the figures/table’s captions (title in bold, legend without bold). We have also changed the name of the figure files and reuploaded them to the platform. 

Reply: Thank you for this relevant comment. We have revised the manuscript and deleted the phrase “data not shown”, which was used twice (pages 15 and 19). We confirmed that, in these two situations, all information is available in the text and there is no need for supplementary data (otherwise we would just be replicating information).

3. Please upload a copy of Supporting Information Figure/Table/etc. "Supporting information" which you refer to in your text on page 35.

Reply: Thank you for pointing this out. In our first submission, we mistakenly listed all tables and figures mentioned in the text in the “Supporting information” section. However, we do not have supplementary content to add to the paper. The health literacy questionnaire under study, correction criteria, and dataset are available on Zenodo (as indicated in the manuscript). In total, we have seven tables (all present in the manuscript file) and three figures (uploaded separately to the submission platform). We have reuploaded all figures because changes were made in the title and legend, following PLOS ONE guidelines.

Reply: Thank you for bringing this to our attention. We have reviewed the reference list and replaced some of them with relevant and more recent references. After this review, we confirmed that there are no retracted papers cited in the manuscript. The changes in the reference list were as follows:

• We have replaced the reference “World Health Organization. Health Promotion Glossary. Geneva; 1998” with 

“Sørensen K, Van den Broucke S, Fullam J, Doyle G, Pelikan J, Slonska Z, et al. Health literacy and public health: A systematic review and integration of definitions and models. BMC Public Health. 2012;12: 80. doi:10.1186/1471-2458-12-80”

• We added a reference in the contextualization of health literacy

“Nutbeam D, McGill B, Premkumar P. Improving health literacy in community populations: a review of progress. Health Promot Int. 2018;33: 901–911. doi:10.1093/heapro/dax015”

• We have replaced the references “White S. Relationship of Preventive Health Practices and Health Literacy: A National Study. Am J Health Behav. 2008;32. doi:10.5993/AJHB.32.3.1” and “Berkman ND, Sheridan SL, Donahue KE, Halpern DJ, Crotty K. Low Health Literacy and Health Outcomes: An Updated Systematic Review. Ann Intern Med. 2011;155: 97. doi:10.7326/0003-4819-155-2-201107190-00005” with 

“Shahid R, Shoker M, Chu LM, Frehlick R, Ward H, Pahwa P. Impact of low health literacy on patients’ health outcomes: a multicenter cohort study. BMC Health Serv Res. 2022;22. doi:10.1186/s12913-022-08527-9”

• We replaced the reference “Baker DW, Wolf MS, Feinglass J, Thompson JA. Health Literacy, Cognitive Abilities, and Mortality Among Elderly Persons. J Gen Intern Med. 2008;23: 723–726. doi:10.1007/s11606-008-0566-4” with

“Fan Z ya, Yang Y, Zhang F. Association between health literacy and mortality: a systematic review and meta-analysis. Archives of Public Health. 2021. doi:10.1186/s13690-021-00648-7”

• We removed the reference “Yan T, Tourangeau R. Fast times and easy questions: the effects of age, experience and question complexity on web survey response times. Appl Cogn Psychol. 2008;22: 51–68. doi:10.1002/acp.1331”, in response to the reviewer's suggestion to streamline the text and eliminate irrelevant information. During our review, the sentence citing this reference was deemed unnecessary and has been deleted.

• We removed the reference “Martins A, Andrade I. Adaptação cultural e validação da versão portuguesa de Newest Vital Sign. Revista de Enfermagem Referência. 2014;IV Série: 75–83. doi:10.12707/RIII1399” because it was the same as the reference as “Martins A, Andrade I. Cross-cultural adaptation and validation of the portuguese version of the Newest Vital Sign. Revista de Enfermagem Referência. 2014;IV: 75–83. doi:10.12707/RIII1399”.

• When exploring the rationale for using exploratory factor analysis a reference was added: “Watkins MW. Exploratory Factor Analysis: A Guide to Best Practice. Journal of Black Psychology. 2018;44: 219–246. doi:10.1177/0095798418771807”.

---

## [Editor Report · Decision Letter 1]

24 Jun 2024

Psychometric properties of the Functional Literacy Questionnaire among Portuguese adolescents

PONE-D-24-07888R1

Dear Dr. Raquel Martins

We’re pleased to inform you that your manuscript has been judged scientifically suitable for publication and will be formally accepted for publication once it meets all outstanding technical requirements.

Kind regards,

Maria José Nogueira, Ph.D.

Academic Editor

PLOS ONE

Additional Editor Comments (optional):

The authors reformulated the manuscript according to the reviewers' recommendations, which greatly added clarity and robustness,

The manuscript is now ready to be accepted for publication.
---

## [Editor Report · Acceptance letter]

14 Aug 2024

PONE-D-24-07888R1 

PLOS ONE

Dear Dr. Martins, 

I'm pleased to inform you that your manuscript has been deemed suitable for publication in PLOS ONE. Congratulations! Your manuscript is now being handed over to our production team.

Kind regards, 

on behalf of

Professor Maria José Nogueira 

Academic Editor

PLOS ONE